

Monitoring Urban Heat Island Intensity with Ground-based GNSS Observations and
Space-based Radio Occultation and Radiosonde Historical Data
**Pengfei Xia[1], Wei Peng[2],Shirong Ye[1], Min Guo[3], Fangxin Hu[1]**
[1]GNSS Research Center, Wuhan University, Wuhan 430079, China
[2]Faculty of Engineering and applied science,University of Regina, Regina, Saskatchewan,
Canada
[3] School of Surveying and Land Information Engineering, Henan Polytechnic University, Jiaozuo
454000, China
*Correspondence to:* Pengfei Xia( xpf130@163.com )
**Abstract.** Since Urban Heat Islands (UHI) not only negatively impact human health but consume
more energy when cooling buildings, accurate monitoring of its impact is critical. In this study, we
propose a ground based GNSS technique to fuse GNSS Radio Occultation (RO) and radiosonde
products to monitor the UHI intensity, which described as follows: First, the first and second grid
tops are defined using the historical RO and radiosonde products. Then, the wet refractivity between
the first and second grid tops is fitted to the higher-order spherical harmonic function based on the
RO and radiosonde products, and they are used as the inputs of GNSS tomography, which can reduce
the number of unknowns voxels of tomography while increasing the effective number of satellite
rays, and improving the accuracy of tomography results. Next, according to the relationships among
wet refractivity, temperature, and water vapor partial, as well as the function relationships among
temperature, wet pressure, and height in adjacent vertical layers, the temperature and water vapor
partial pressure can be obtained using the best search method according to the tomography-derived
wet refractivity. Finally, the UHI intensity is monitored by the temperature difference between the
urban regions and the surrounding rural regions. The radio occultation and radiosonde products of
the Hong Kong region from 2010 to 2019, and the observed GNSS network data of the Hong Kong
region for the year of 2020 are employed to evaluate the UHI intensity algorithm. The validation of
the algorithm is done by comparing the UHI intensity estimated from the algorithm with the
temperature data obtained from weather stations. The result shows that the proposed algorithm can
achieve an accuracy of 1.2 K at a 95% confidence level.
**Keywords:** Urban heat island (UHI), GNSS, Tomography, Temperature
**1. Introduction**
The urban heat island (UHI) effect arises as urban regions become warmer than their rural
environments (Roth, 2013). The UHI is mainly caused by heat absorbed by built structures and
anthropogenic heat sources in cities (Roth, 2013). The UHI intensity is related to many factors such
as regional climate, urbanization, and topography, etc. Since the 21st century, with the rapid
development of urbanization, densified urban building clusters, and a large number of people
gathering in cities, the increases in industrial production and domestic energy use have intensified
the UHI effect, leading to a continuous expansion of its scope and intensity (Zhai et al., 2018; Xu et
al., 2018; Zou et al., 2019; Jiang et al., 2019). The heat island effect severely affects the life of
citizens and even the overall ecological quality of cities, because it increases the heat stress of
citizens, which further triggers cardiovascular, respiratory and mental diseases, resulting in
increased morbidity and mortality worldwide. The study of the urban heat island formation and
evolution, spatial and temporal distribution, causal mechanisms and the search for effective
mitigation measures have become the focus of attention of many scholars in recent years (Rizwan



et al., 2008; Memon et al., 2009; Azevedo and Leal, 2017; Lamarca et al., 2018).
Traditionally, three techniques are used to monitor the UHI intensity: a network of ground-based
temperature sensors (Ramamurthy and Sangobanwo, 2016); remote Sensing Satellite data (Schwarz
et al., 2011; Wu et al., 2014; Fang et al., 2016; Kayet et al., 2016) and airborne instruments (Peng
et al., 2017). The above observation methods have some drawbacks such as low spatial resolutions,
high cost and weather dependence (Jorge et al., 2021). For example, when temperature sensors are
used to obtain the relevant patterns, it is mainly to compare the temperatures of urban and suburban
areas during the same period, but due to the limited monitoring points, it is difficult to reflect the
UHI effect of the study area comprehensively. To address this issue, a wide metropolitan area has
been covered by a dense sensor network, which no doubt leads to increased monitoring costs
(Jauregui, 1997; Jin, 2012). Satellite imaging and airborne instruments require clear-sky conditions
to obtain accurate data (Grimmond et al., 2010; Vahmani and Ban-Weiss, 2016). GNSS (Global
Navigation Satellite Systems) technology, a new means of atmospheric sounding, can not only
effectively overcome the shortcomings of traditional means (i.e., can compensate for these
disadvantages), but also has the advantages of high observation accuracy, quasi-real-time, all-
weather, no need for human interference, no need for instrument calibration, etc (Kouba and Héroux,
2001; Cai et al., 2013 and 2015).
A novel method of monitoring the UHI intensity using GNSS data was first presented by Jorge
et al (2021). This algorithm is based on the relationship between the single GNSS-derived Zenith
Tropospheric Delay (ZTD) and the environmental variables (pressure, water vapor partial pressure,
and temperature) at the measurement site. The UHI intensity is calculated by subtracting the
temperature at an urban GNSS station from the temperature at a rural GNSS station. However, due
to the limited number of GNSS stations in each city, the single-station GNSS inversion of
atmospheric temperature cannot be used for the heat island effect in urban areas. In recent years, the
GNSS tomography technique has been applied as an effective means to acquire the three-
dimensional (3-D) distribution of wet refractivity and can compensate for single GNSS algorithm's
disadvantages. In order to monitor the UHI intensity, it is necessary to estimate the temperature from
GNSS-derived wet refractivity (Troller et al., 2006; Bender et al., 2011; Lutz et al., 2010; Rohm,
2013; Chen et al., 2014; Xia et al.,2018). Therefore, the quality of the wet refractivity determines
the accuracy of the temperature inversion.
The key advantage of the GNSS tomography technique over the single GNSS method is that it
can obtain the three-dimensional distribution of the temperature in the study area, and can study the
temperature changes in the horizontal direction and vertical direction. The second advantage is that
it can promote the application scope of GNSS sensors. The proposed paper developed an optimized
method for GNSS 3-D troposphere tomography using the external radiosonde and GNSS radio
occultation (RO) historical data. Among them, the radiosonde and RO products are utilized to
determine the first grid top and the second grid top for the purpose of grid division. Further, the wet
refractivity is obtained between the first grid top and the second grid based on radiosonde and RO
data which are used as the input value of the GNSS tomography. Next, the temperature can be
obtained using the optical search method from tomography-derived wet refractivity. The ground-
based GNSS observation data from the Hong Kong SatRef network in 2020 are used to verify the
feasibility and superiority of the optimized tomography method. In addition, the temperature from
5 weather stations in Hong Kong is selected as a reference to validate the temperatures obtained by
GNSS data.





The paper is organized as follows: In section 2, the method development is presented. Then,
Section 3 describes the processing of the data, before the discussions in Section 4, followed by the
conclusions in Section 5.
**2.    Methodology**
This section introduces the basic tomography model; describes the tomography grid division and
the calculation of temperature from wet refractivity; and presents the calculations of the UHI
intensity from temperature.
*2.1 Tomography model*
The SWD along the ray paths traversing the imaging region should first be derived from the dual-
frequency GNSS data to reconstruct the 3D images of the atmospheric wet refractivity distributions,
which is defined by the line integral of $N_w$ along the ray path from the satellite to the receiver (Flores
et al., 2001) as follows:
$$SWD = 10^{-6} \cdot \int_{h_0}^{\infty} N_w \cdot dh \qquad (1)$$
$$SWD = 10^{-6} \cdot \sum_{i=1}^{m} \sum_{j=0}^{4} C_{i,j}^4 \cdot N_w(s_j^i) = S \cdot N_w + \Delta_{SWD} \qquad (2)$$
where, $C_{i,j}^4$ is the 4-order coefficients of the j-th segment point within the i-th grid; $N_w(s_j^i)$ is the
corresponding atmospheric wet refractivity; $m$ denotes the number of the grids which the signals had
passed; $S$ represents the distance of the GNSS signals spanning the voxel, and $\Delta_{swv}$ is the noise.
Due to the near-cone geometry of GNSS observations (Bender and Raabe, 2007; Benevides et al.,
2016), the GNSS signal cannot pass through all voxels, which resulted in too many zeros in the
design matrix, and the tomographic system cannot be inverted. Therefore, it is necessary to apply
appropriate constraints to overcome this issue. Though the Gauss weighted method (Song, 2004) can
be used for the horizontal direction, the vertical distribution is still modeled by an exponential
equation, taking account of the water vapor in the vertical direction ( which usually decreases with
increasing height) as follows:
$$N_w(h) = N_C \cdot e^{\left(-\frac{(h-h_0)}{H_z}\right)} \qquad (3)$$
where, $N_w(h)$ denotes the atmospheric wet refractivity at the height of $h$; $H_z$ represents the height
index of $N_w$; $N_C$ is the constant value; $h_0$ is a constant. Based on Eq. (3), Eq. (4) is employed as the
vertical constraint to establish the relationship between atmospheric wet refractivity in adjacent
vertical layers:
$$\frac{N_w^{i,j,k+1}}{N_w^{i,j,k}} = e^{\left(\frac{h_k-h_{k+1}}{H_z}\right)} \qquad (4)$$
where, the subscripts "$i$", "$j$", and "$k$" define the indexes of the voxels in the east-west, north-south,
and vertical directions, respectively; $h_k$ is the height of the $k^{th}$ voxel. Thus, the tomography equation
can be solved by adopting Kalman filtering based on the horizontal and vertical constraints.
*2.2 Tomography grid division*
Generally, both the lower and upper limits of the tomographic grid refer to the height from the
ground to the top of the tropopause. However, the wet refractivity is mostly clustered at a height that
significantly below the tropopause. If the top of grid uses tropopause, the solutions of the
tomography inversion may be negative since the wet refractivity is very sparse near the height of the
tropopause (Flores et al., 2000). Two grid heights in this study are defined according to the ZWD
variations obtained from radiosonde and Radio occultation data. The first grid top is the upper limit





of the tomography grid, but when the tomography equation cannot calculate the $N_w$ value between
the first and the second grid tops, the radiosonde-derived $N_w$ or RO-derived $N_w$ are used to take its
place. As a result, when the height of the grid top decreases, the effective number of satellite rays
increases. In the tomography, the rays penetrating the grid from the top boundary are the only rays
used in the tomographic solution.
GNSS tomography aims to reconstruct the vertical distribution of $N_w$. The division of the vertical
grid severely affects the tomography solutions. Conventionally, two approaches have been used for
dividing the grid: the uniform division (Flores et al., 2000; Xia et al., 2013 and 2018) and non-
uniform division (Perler et al., 2011; Rohm, 2012 and Jiang et al., 2014). Considering the practical
distribution characteristics of $N_w$ are sparse in high layers and dense in low layers, the non-uniform
division is used here.
*2.3  Calculation of temperature from $N_w$*
The ZWD is related to the environmental conditions because of the wet refractivity of the
troposphere. The wet refractivity $N_w$ of the troposphere is defined as:
$$N_w = \left(k_2 - k_1 \frac{R_d}{R_w}\right) \cdot \frac{P_w}{T} + k_3 \cdot \frac{P_w}{T^2} \tag{5}$$

where, the empirically calculated constants $k_1$=77.6, $k_2$=72, and $k_3$=3.75×10$^5$; $R_d$ and $R_w$ are
mean specific gas constant for day air and water vapor, respectively; $P_w$ is the water vapor partial
pressure, and $T$ is the temperature in Kelvins.
The temperature of the grid point is calculated by putting zero to the equation of wet refractivity
(Eq.5) and solving the quadratic equation:
$$T^2 \cdot N_w - T \cdot \left(k_2 - k_1 \frac{R_d}{R_w}\right) \cdot P_w - k_3 \cdot P_w = 0 \tag{6}$$

Although the $N_w$ can be obtained from the above tomography equation, the $P_w$ and $T$ cannot be
solved directly because the equation (6) is rank deficient. Therefore, Eq. (6) requires additional
conditions to calculate $P_w$ and $T$.
In the vertical direction, the temperature decreases with height at a relatively consistent rate and
denoting the lapse rate $\beta$, we have
$$T_{i+1} = T_i - \beta \cdot (h_{i+1} - h_i) \tag{7}$$

where, $T_i$, $T_{i+1}$ represent the temperature at the $i_{th}$ and $(i+1)^{th}$ grids, respectively; $h_i$, $h_{i+1}$
represent the height at the $i^{th}$ and $(i+1)^{th}$ grids, respectively.
The water vapor partial pressure usually decreases with height, and it can be expressed as an
empirical exponential function (Callahan, 1973):
$$P_w^{i+1} = P_w^i \cdot \exp(a \cdot (h_{i+1} - h_i) - b \cdot (h_{i+1} - h_i)^2) \tag{8}$$

where, $P_w^i$, $P_w^{i+1}$ represent the water vapor partial pressure at the $i_{th}$ and $(i+1)^{th}$ grids, respectively;
$a$ and $b$ are constant, which can be obtained based on radiosonde and RO products.
Combining Equations (6), (7) and (8), the search ranges for water vapor partial pressure and
temperature are given based on radiosonde data, and then the optimal water vapor partial pressure
and temperature are searched.
*2.4 Calculation of the UHI intensity*
The UHII can be calculated by estimating the temperature difference between urban and rural grid
points based on ground-based GNSS tomography technique. Eq. (9) shows the calculations:



$$UHII_{GNSS} = T_{GNSS}(urban) - T_{GNSS}(rural) \qquad (9)$$

where, $T_{GNSS}$ is the temperature in Kelvins obtained from GNSS tomography. The algorithm
usually can be validated by comparing the UHII obtained using GNSS data with the UHII obtained
using meteorological data.
$$UHII_{Met} = T_{Met}(urban) - T_{Met}(rural) \qquad (10)$$

where, $T_{Met}$ is the temperature obtained from meteorological data.
**3 Processing results and analysis**
*3.1 Data description*
Since 2001, there has been a substantial supply of continuous temperature and pressure
measurements provided by the satellite-based GNSS RO data with high accuracy, high vertical
resolution, and global coverage as a function of altitude in the upper troposphere and lower
stratosphere. In this paper, we used the most recent Wegener Center (WEGC) multi-satellite GNSS
RO data, OPSv5.6, from May 2001 to December 2020. WEGC OPSv5.6 has been widely used for
weather, climate, space weather, and geodetic studies as it provides global upper-air satellite data in
high quality from multiple RO satellite missions, which including CHAMP, GRACE, SAC-C,
Formosat-3/COSMIC, and Metop (Schreiner et al., 2007; Anthes et al. 2008). The radiosonde
technique is a tool for studying meteorology from the ground to the lower stratosphere. RO and
radiosonde observations are key data sources for weather studies and climate analysis (Kuo et al.,
2005; Kishore et al., 2011).
The Ground-based GNSS observation data are obtained from the Hong Kong Satellite Navigation
System (HKSN) network, which is composed of 12 continuously operating stations with a distance
of 10-15 km between stations, as shown in Fig. 1. All 18 stations are provided by "LEICA
GRX1200+GNSS" receivers with a data sampling rate of 5s. The 2020 one-year GNSS dataset is
collected in Hong Kong. Additionally, the RO wet profiles with the same Hong Kong address and
the radiosonde products at the "45004th" station from 2010 to 2019 are utilized as historical data
for optimization of tomographic solutions.

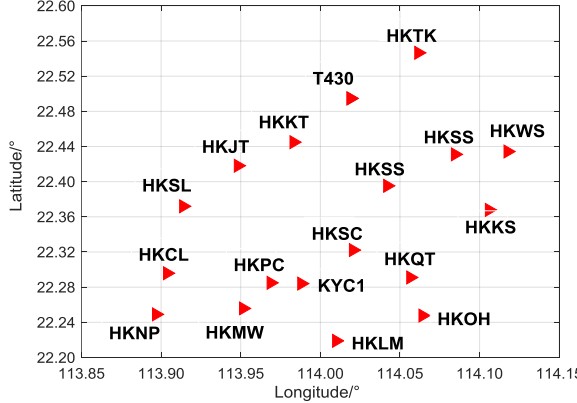

**Fig.1.** The GNSS station distribution and horizontal grid division in Hong Kong

A sliding time window strategy is a great approach for the simulation of near real-time GNSS
tomographic experiment (Foster et al., 2005). Furthermore, providing a logical time interval is




meaningful in the framework of the rain now-casting. Moreover, a 6h time interval for the minimum
broken line length is recommended empirically to allow the linear fitting algorithm can conduct a
better discretization of the PWV signal features without being affected by the noisy features
(Benevides et al., 2015). Therefore, a six-hour interval's time window is used, moving forward by
an hour each time. The GAMIT software is used in this study to obtain ZTD (Herring et al., 2010).
### 3.2 Defining the grid top
In the zenith direction, the wet tropospheric delay can be expressed as:
$$\mathrm{ZWD}_h = 10^{-6} \int_h^\infty N_w \cdot d_h \tag{11}$$

where, $\mathrm{ZWD}_h$ is the wet tropospheric delay (unit: m); $h$ is the height of the observation station
over mean sea level (unit: m); $N_w$ is the atmospheric wet refractivity (unitless) that can be modeled
from Eq.(5).
In this study, the ZWD is obtained from radiosonde data and RO profiles. They are used to define
the grid top. The radiosonde sensors can measure several meteorological parameters such as
pressure, temperature, and relative humidity. Similar to radiosonde, RO profiles also provide
meteorological products such as temperature, water vapor pressure, etc. Taking the characteristic of
exponential decreasing of the atmospheric refractivity into account, the formulas for wet delays
from the radiosonde measurements and RO profiles can be derived:
$$\mathrm{ZWD}_h = 10^{-6} \sum_i \left[ (h_i - h_{i+1}) \left( N_w^{i+1} - N_w^i \right) / \left( \ln N_w^i - \ln N_w^{i+1} \right) \right] \tag{12}$$

Afterward, slant wet delay (SWD) can be obtained from ZWD based on the wet Niell mapping
function (Niell, 1996).
$$\mathrm{SWD}_h = \mathrm{ZWD}_h \cdot M_w^{\mathrm{Niell}}(e_{min}) \tag{13}$$

where, $M_w^{\mathrm{Niell}}$ is the wet Niell mapping function; $e_{min}$ means the minimum satellite cut-off angle,
which is 10 °in this study. When the $\mathrm{SWD}_h$ is less than or equal to 1 mm, the corresponding height
is defined as the first grid top (FGT). The difference between ZWD and $N_w$ between two adjacent
time periods can be calculated:
$$\Delta\mathrm{ZWD}_h^t = \left| \mathrm{ZWD}_h^t - \mathrm{ZWD}_h^{t+1} \right| \tag{14}$$

$$\mathrm{RMSN} = \sqrt{\frac{\sum_{i=1}^n \left( N_w^{h_i,t} - N_w^{h_i,t+1} \right)^2}{n}} \tag{15}$$

where, $\mathrm{ZWD}_h^t$ and $\mathrm{ZWD}_h^{t+1}$ are the ZWD at height $h$ at time $t$ and $t$+1, respectively. $N_w^{h_i,t}$ and
$N_w^{h_i,t+1}$ are the height $h$ at time $t$ and $t$+1; $n$ is the number of the layers of radiosonde and RO data.
When the $\Delta\mathrm{ZWD}_h^t$ is less than or equal to 0.5 mm or RMSN is less than or equal to 0.5N, the
corresponding height is defined as the second grid top (SGT).
RO wet profile and radiosonde products at the "45004th" station from 2010 to 2019 are used to
determine the grid top based on Equations (13), (14) and (15). Individual radiosonde data, which
have different vertical resolutions, are linearly interpolated to a 100-m vertical grid before the grid
top identification. Then, we compute the daily mean for the first grid top and the second grid top as
shown in Figure 2.



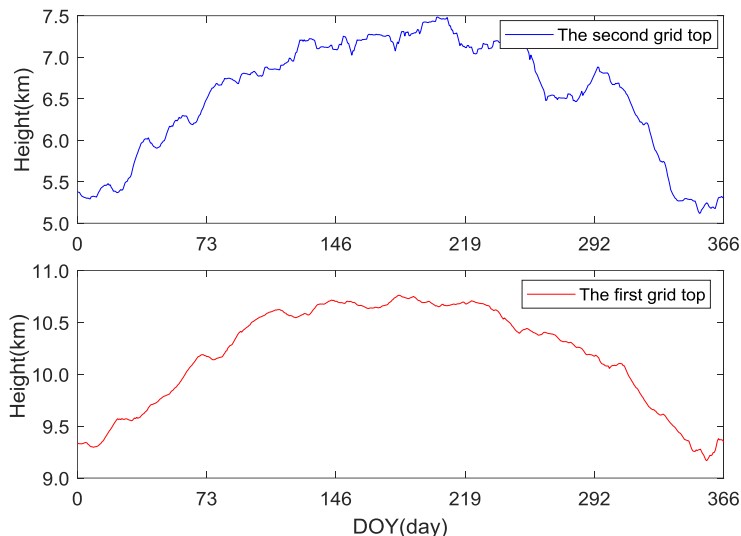


Figure 2. The first grid top and the second grid top obtained by radiosonde and RO products.

It can be seen from Figure 2 that FGT and SGT are significantly higher in summer than in spring
and winter. Compared with spring and winter, the SGT is 1-2 km higher in summer than in spring
and winter, while the FGT is 0.5-1.5 km higher in summer than in spring and winter. The variation
of the two grid heights with time is a non-smooth curve, and the SGT is more jittered than the first
grid height. Thus, we divide the vertical layers into three stages. The first stage is from the ground
to 1km, which be further divided into 3 layers: the heights of the first two layers are 300m, and the
height of the third layer is 400m. The second stage is from 1km to SGT. The grid in this stage is
divided into an even vertical height which requires the height of a grid is no less than 400m and not
more than 600m. The last stage is from the SGT to the FGT. The grid in this stage is divided into an
even vertical height which requires the height of the grid is no less than 600m, and not greater than
1000m.
*3.3 Obtaining the $N_w$ between FGT and SGT*
The RO wet profile and Radiosonde product have been quality controlled, and $N_w$ can be obtained
from the water vapor pressure and temperature provided by the two meteorological data in equation
(5). We add daily and semidiurnal terms to the annual and semiannual cycle variation characteristics
of $N_w$, and the $N_w$ time series obtained by the following equation which is layered for periodic fitting
(from 5km to 11km, it is divided into 12 layers on average, that is, a layer of 500m).
$$N_w^j = \sum_{n=0}^2 a_n^j \cos\left(2n\pi \frac{\mathrm{doy}-b_n^j}{365.25}\right) + \sum_{n=3}^4 a_n^j \cos\left(2(n-2)\pi \frac{\mathrm{hod}-b_n^j}{24}\right) \qquad (16)$$
where, $j$ is the number of layers; $N_w^j$ is the wet refractivity of the $j^{th}$ layer; doy is the annual
cumulative day; hod is the UTC time; $a_n$ ($n$=0,1,2) is the annual mean; annual cycle variation
amplitude and semiannual cycle variation amplitude of $N_w$, $b_n$ ($n$=1,2) is the annual cycle variation
initial phase and semiannual cycle variation initial phase; $a_n$ ($n$=3,4) is the daily cycle variation
amplitude and semiannual cycle variation amplitude; $b_n$ ($n$=3,4) is the daily cycle variation initial
phase and semiannual cycle variation initial phase, respectively. The values of $a_n$ and $b_n$ in Eq.(16)



at different altitude levels can be fitted by selecting the RO and radiosonde products in Hong Kong
from 2010 to 2019.
SWD is the input value of the GNSS tomography technique, and its accuracy directly affects the
accuracy of tomography-derived $N_w$. To evaluate the fitting accuracy of $N_w$ between SGT and FGT,
we selected the 2020 Hong Kong area radiosonde and RO products as the benchmark values. The
difference of SWD can be obtained between benchmark value and $N_w$ for different seasons derived
from Eq. (16), as shown in Eq. (17). The statistical results are displayed in Table 1.

$$\Delta \text{SWD}_{SGP}^{FGP} = (\text{ZWD}_{model} - \text{ZWD}_T) \cdot M_w^{\text{Niell}}(e) \qquad (17)$$

where, $\text{ZWD}_{model}$ denotes the ZWD obtained from Eq. (5) and Eq. (16); $\text{ZWD}_T$ denotes the
ZWD estimated from RO and radiosonde products using Eq. (5); $e$ is the satellite cut-off angle;
$M_w^{\text{Niell}}$ is the Niell wet projection function.

**Table 1.** Results of the differences of SWD between model-derived and

benchmark values at different cut-off angles (Unit mm).

| Cut-off angle | Spring | Summer | Autumn | Winter |
|---|---|---|---|---|
| 7°-15° | 3.6 | 5.1 | 2.7 | 2.2 |
| 15°-30° | 1.5 | 2.2 | 1.4 | 1.2 |
| 30°-45° | 0.9 | 1.2 | 0.9 | 0.8 |
| 45°-60° | 0.7 | 0.9 | 0.6 | 0.6 |
| 60°-75° | 0.6 | 0.8 | 0.6 | 0.5 |
| 75°-90° | 0.5 | 0.7 | 0.5 | 0.5 |

Table 1 shows that the value of $\Delta \text{SWD}_{SGP}^{FGP}$ is significantly larger at lower satellite cut-off angles.
In addition, $\Delta \text{SWD}_{SGP}^{FGP}$ is significantly larger in summer than in other seasons, with a deviation of
more than 5mm. At satellite cut-off angles above 45°, the effect of $\Delta \text{SWD}_{SGP}^{FGP}$ is less than 1mm,
while at satellite cut-off angles below 30°, the effect of $\Delta \text{SWD}_{SGP}^{FGP}$ is greater than 1mm.
**4. Discussion**
The 450045[th] radiosonde station was carried aloft once every 12h in Hong Kong, and it was equipped
with a configured sensor that collects information about temperature, pressure, relative humidity and
so on. Here, the 450045[th] radiosonde products in 2020 were used to evaluate the accuracy of the
tomography results. These products were further used to validate the accuracy of the temperature
obtained from $N_w$.
*4.1 GNSS tomographic results*
We utilized GAMIT software to obtain the ZTD based on GNSS data from the Hong Kong SatRef
network with IGS (International GNSS Service) ultra-rapid products orbit file. The Saastamoinen
model and GPT3 model were used to obtain the ZHD and then the ZWD was obtained by deducting
the ZHD from the ZTD. The SWD was computed from ZWD using Niell wet mapping function.
Finally, we estimated the 3-D distribution of atmospheric wet refractivity using parameterized
approaches, in which Eq. (16) was used for deriving the $N_w$ between SGT and FGT. Besides, Kalman
filtering algorithm was used for tomography solutions.
To evaluate our optimized method, the tomography results were compared with those derived
from the traditional tomography method, in which the atmospheric wet refractivity from SGT to
FGT was estimated as unknown. First, the numbers of signals passing through the voxel (NSV) were
compared when the optimized method and traditional method were used to invert the $N_w$ given in
Figure 3. Then, tomography solutions were compared with external results derived from the





radiosonde. The results are shown in Figure 3.

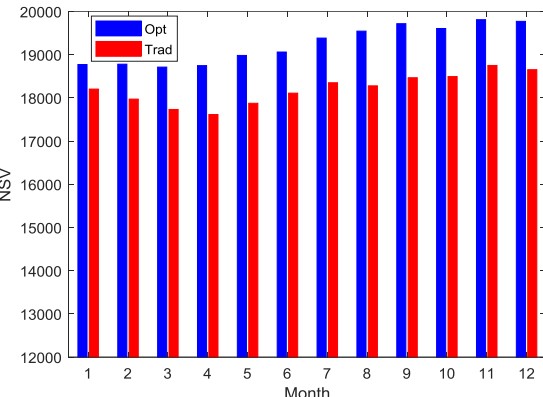


**Figure 3** Comparison results of the number of signals passing through voxel between the traditional method and
the optimized method. Opt refers to the optimized method; Trad refers to the traditional method.


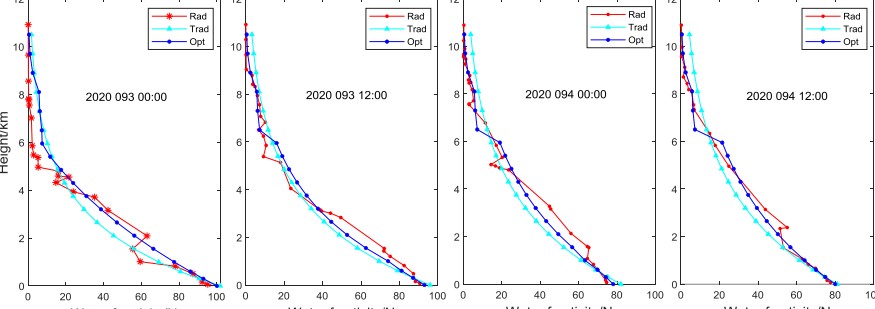


**Figure 4**. Wet refractivity obtained from tomography-derived and radiosonde-derived data. Rad is the wet
refractivity derived using radiosonde products, Trad is the wet refractivity derived using the traditional tomography
method, and Opt is the wet refractivity derived using the optimized method.

As shown in Fig.3, the average number of NSVs per month of the optimized method is higher
than that of the traditional method. From August to December, the average monthly NSVs of the
optimized tomography is more than a thousand signals than the traditional tomography. Statistics
show that the NSVs in the optimized technique is 5.8% better than that of the traditional technique.
As can be seen in Fig. 4, there is a good agreement between the changing trends of wet refractivity
with height across the tomography-obtained and data from radiosonde. However, in the case when
the "inversion layer" is present, GNSS tomography fails to accurately represent in this situation. The
wet refractivity derived from our optimized method is better than that from the traditional method
since the blue curve is closer to the red curve. In Table 2, we present the deviation statistics for GNSS
tomography-obtained and radiosonde-obtained wet refractivity over the whole year 2016.
**Table 2**. The statistical results between tomography-derived and radiosonde-derived wet refractivity (Unit: N).

| Season | Rad-Trad | | | | Rad-Opt | | | |
|---|---|---|---|---|---|---|---|---|
| | Max | Min | Mean | RMS | Max | Min | Mean | RMS |





| | | | | | | | |
|---|---|---|---|---|---|---|---|
| Spring | 10.53 | -20.31 | -3.92 | 7.66 | 9.21 | -16.36 | -2.31 | 5.89 |
| Summer | 19.70 | -23.53 | -4.82 | 10.17 | 14.66 | -18.54 | -3.56 | 8.14 |
| Autumn | 11.15 | -21.05 | -3.09 | 9.03 | 9.95 | -17.24 | -3.05 | 7.82 |
| Winter | 10.73 | -13.77 | 0.46 | 5.64 | 9.15 | -11.95 | -1.25 | 5.03 |


Table 2 provides statistical values of the differences between GNSS tomography-obtained and
radiosonde-obtained results. As seen from the statistical results, the root mean square (RMS) and
mean values of troposphere tomography using the optimized technique is less than that of the
traditional method. Especially in summer, the optimized method is slightly better than other seasons.
In addition, compared with the radiosonde data, the test results show that the wet refractivity quality
obtained by the optimized technique is 16.8% better than that of the traditional technique.
*4.2 Validation of temperature results*
After obtaining the wet refractivity profile based on the GNSS tomography method, the temperature
was estimated by the optimal search method using eqations (6), (7) and (8). The fifth-generation
reanalysis model (ERA5) could provide temperature and water vapor partial pressure, which were
selected as the initial values in this study. Since the temperature and water vapor pressure provided
by ERA5 are inconsistent with the spatial and temporal resolution of the tomographic results, the
Gaussian distance weighting function in the horizontal direction and the exponential function in the
vertical direction are used to interpolate ERA5 to be consistent with them. In terms of time, the
temperature and water vapor partial pressure of ERA5 can be interpolated by the Chebyshev
function of order 9, which can achieve a time resolution consistent with the tomography results.
Since our research area is Hong Kong, China, and the tallest building in this area is not more than
600m, we only calculated the temperature at the vertices of each grid layer below 600m. If
determining the appropriate search range, it is crucial to find the range of percentage deviation
between benchmark value and ERA5 product. Then, using the radiosonde product as the benchmark
value, calculate the difference between the temperature and water vapor partial pressure provided
by ERA5 and the radiosonde product below 600m. This deviation can be formulated as follows:
$$DT = \frac{RADT - ERAT}{RadT} \cdot 100 \tag{18}$$

$$DWP = \frac{RADWP - ERAWP}{RadWP} \cdot 100 \tag{19}$$

where, the RADT and RADWP are the temperature and pressure provided by radiosonde,
respectively, and the ERAT and ERAWP are the temperature and pressure provided by ERA5,
respectively. To study the range of percentage deviation of DT and DWP, we computed the situation
in Hong Kong from 2010 to 2019 based on Eqations (18) and (19).

**Table 3.** Summary of the change intervals of temperature and water vapor
pressure between ERA5 and Radiosonde from 2010 and 2019.

| DT | | | DWP | | |
|---|---|---|---|---|---|
| [-0.75%,0.75%] | [-1%,1%] | [-1.5%,1.5%] | [-7.5%,7.5%] | [-10%,10%] | [-15%,15%] |
| 64.5% | 77.7% | 93.5% | 46.5% | 59.2% | 71.4% |

Table 3 provides the statistics on the scope of DT and DWP in Hong Kong. If the ranges of DT and
DWP are too large, some of the temperature and water vapor partial pressure may be over-corrected,
but if the range of DN is too small, the temperature and water vapor partial pressure may be under-





corrected. In this study, [-0.75%,0.75%] in temperature and [-10%,10%] in water vapor partial
pressure are selected as the range of theoretical retrieval. Then, the theoretically retrieved value of
atmospheric temperature is obtained at each layer as CT + CT·DT, where the search step size of DN
is 0.25%. The theoretically retrieved value of atmospheric water vapor partial pressure is obtained
at each layer as CWP + CWP·DWP, where the search step size of DN is 2.5%. Finally, the optical
CT+CT·DT values are derived based on the equations (6), (7) and (8). Figure 5 gives the 3-D
temperature distribution on Hong Kong below 600m on April 2 and 3, 2020.

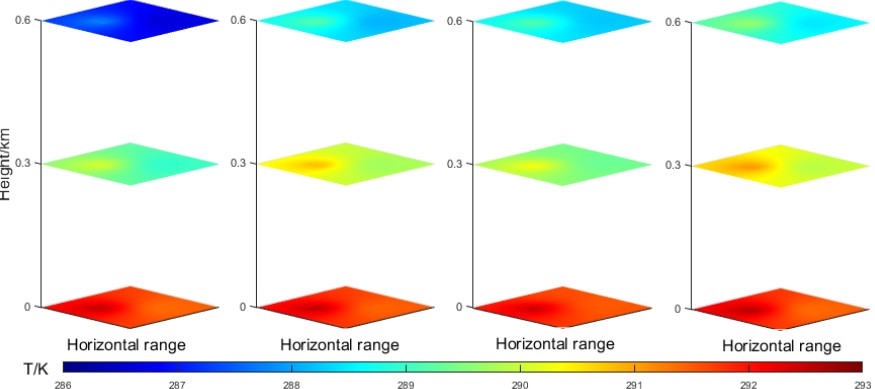


**Figure 5.** 3-D distribution of atmospheric temperature below 600m on April 2 and 3, 2020.
Figure 5 describes the water vapor density changes at different heights. It shows that the
atmospheric temperature tends to decrease significantly with elevation. In the horizontal direction,
the temperature of the first layer does not change significantly over time, while the temperature of
the second and third layers changes more obviously over time. In order to verify the accuracy of the
inversion results of the temperature and water vapor pressure, we selected the radiosonde products
in 2020 as the true value, and compared them with the inversion results corresponding to time and
space. The statistical results are shown in Table 4:
**Table 4.** Statistical results between GNSS-derived and radiosonde-derived
temperature and water vapor partial pressure below 600m.

| Season | datT (K) | | | | datWV(hPa) | | | |
|---|---|---|---|---|---|---|---|---|
| | Max | Min | Mean | RMS | Max | Min | Mean | RMS |
| Spring | 3.65 | -2.17 | 0.39 | 1.32 | 3.66 | -2.52 | 0.72 | 1.53 |
| Summer | 2.56 | -2.75 | -0.46 | 1.65 | 4.12 | -3.25 | 0.48 | 1.98 |
| Autumn | 1.91 | -2.52 | -0.54 | 0.96 | 3.54 | -2.54 | -0.67 | 1.47 |
| Winter | 3.28 | -1.94 | 0.82 | 1.45 | 3.05 | -2.33 | 1.13 | 1.31 |


Table 4 provides the different maxima, minima, means and RMSs of GNSS-derived and
radiosonde-derived temperatures and water vapor partial pressures. In terms of the statistical results,
the accuracy of GNSS-derived temperature and water vapor partial pressure in autumn is better than
other seasons. In addition, the best statistical accuracy of GNSS-derived water vapor partial pressure
is in winter while the worst is in summer. This can be attributed to summer and winter usually being
the most and least humid seasons of the year, respectively.



*4.3 The urban heat island (UHI)*

The UHI intensity is defined by the difference between the temperature in urban areas and surrounding rural areas. In urban areas, anthropogenic sources of heat are present, such as transportation and air conditioning equipment. In contract, the quantity and variety of anthropogenic heat sources are less in rural areas because there are few existing buildings and most of them are occupied by nature. It is common for rural and urban areas to be interdependent, with rural areas located outside of urban or city areas (Memon et al., 2009). In order to monitor the intensity of the UHI in Hong Kong, we selected several GNSS stations in the urban area (equipped with meteorological observation) as urban stations, and a weather station as a rural station which is located on a surrounding independent island. The distribution of the stations is shown below:

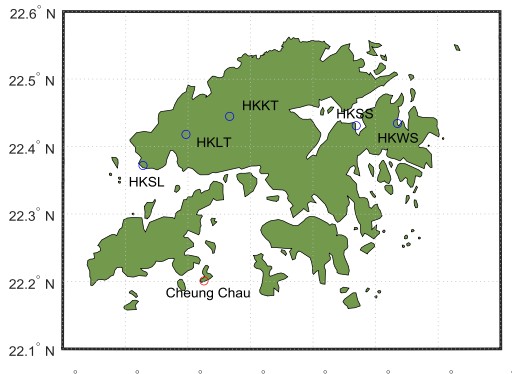

**Figure 6** Distribution of selected GNSS stations and weather stations in Hong Kong. The blue circle indicates the GNSS station, red circle indicates the weather station.

The daily maximum, minimum and average values have been obtained with meteorological data. We fitted these values into a second-order linear function separately. Thus, the maximum, minimum and average values of the UHI intensity in meteorological data were calculated using Eq. (10). In addition, in order to validate the UHI intensity in GNSS data, the temperature obtained by GNSS was interpolated to a same spatial and temporal resolution as the meteorological data using a linear function. Similarly, the maximum, minimum and average of the UHI intensity in GNSS data were calculated using Eq. (9). The results of one of the meteorological stations and the GNSS results that matched with meteorological stations are shown in Figure 7 and Figure 8, respectively.





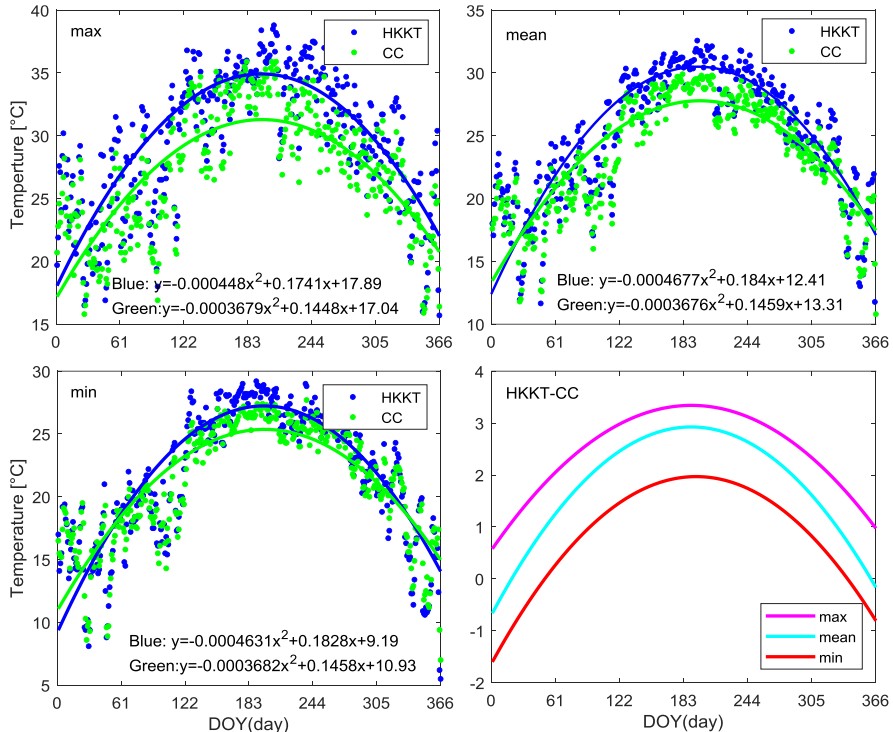

**Figure 7** UHI intensity estimated with meteorological data between HKKT station and CC station. CC refers to
Cheung Chau station.




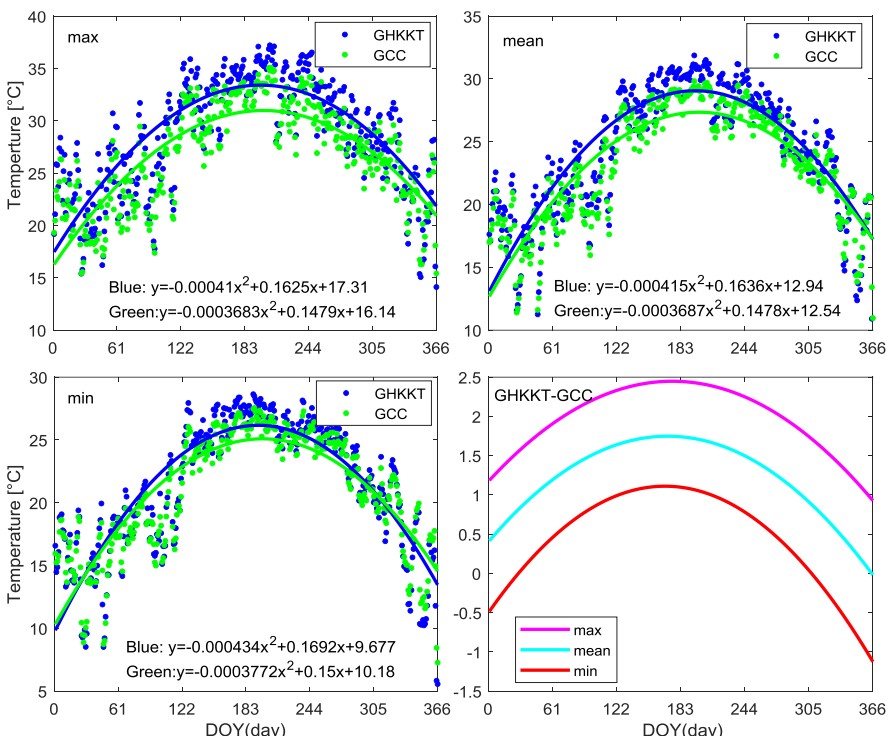

**Figure 8** UHI intensity estimated with ground-based on GNSS observation data. GHKKT refers to the GNSS-derived temperature matched with HKKT meteorological stations. GCC refers to the GNSS-derived temperature matched with Cheung Chau meteorological stations.

The range of the UHI intensity obtained with meteorological data and GNSS data between HKKT and CC is shown in Figure 7 and Figure 8. The shape of the graphs obtained using both data is very similar. In summer, the UHI intensity increases compared to winter. In addition, compared with meteorological data-derived UHI, the UHI obtained from GNSS data is smaller. Beyond that, the 5 pairs of meteorological and GNSS data were used for validation purposes, and the Root Mean Square of the differences between rural and urban areas from meteorological data and GNSS data in different seasons are shown in Table 5 and Table 6. Finally, the validation of the algorithm had been carried out by comparing the UHI intensity (UHII) which determined from GNSS data ($UHII_{GNSS}$) with the UHII which calculated from temperature sensors at weather stations ($UHII_{met}$). The difference in intensity on a given day of the year (Diff_UHII(DOY)) had been compared using the following simple calculation:

$$Diff\_UHII(DOY) = UHII_{GNSS}(DOY) - UHII_{met}(DOY) \qquad (20)$$

The RMS values of the differences of the results obtained from GNSS and from meteorological which using both all data and seasonal data are shown in Table 8. The RMS values of the differences were used to validate the algorithm. The 5 pairs of meteorological and GNSS data used for validation purposes are clearly related to location, as described in Table 7.

**Table 5.** Average UHI intensity from each season obtained using meteorological data in 2020 (Unit: K).

| Pair of stations | 1-year data | Spring | Summer | Autumn | Winter |
|---|---|---|---|---|---|



| | | | | | |
|---|---|---|---|---|---|
| HKKT-CC | 1.74 | 1.99 | 1.94 | 1.64 | 1.36 |
| HKLT-CC | 1.44 | 1.54 | 1.49 | 1.32 | 1.40 |
| HKSL-CC | 1.39 | 1.51 | 1.46 | 1.24 | 1.31 |
| HKSS-CC | 1.37 | 1.44 | 1.41 | 1.22 | 1.39 |
| HKWS-CC | 1.53 | 1.59 | 1.55 | 1.46 | 1.53 |


**Table 6.** Average UHI intensity from each season obtained using GNSS data in 2020 (Unit: K).

| Pair of stations | 1-year data | Spring | Summer | Autumn | Winter |
|---|---|---|---|---|---|
| GHKKT-GCC | 1.22 | 1.33 | 1.28 | 1.19 | 1.22 |
| GHKLT-GCC | 1.06 | 1.18 | 1.05 | 0.96 | 1.06 |
| GHKSL-GCC | 1.11 | 1.19 | 1.13 | 1.01 | 1.06 |
| GHKSS-GCC | 0.93 | 1.05 | 0.92 | 0.85 | 0.87 |
| GHKWS-GCC | 0.98 | 1.08 | 0.96 | 0.82 | 0.97 |


**Table 7.** Relation of meteorological and GNSS pairs.

| | Meteorological data | GNSS data |
|---|---|---|
| UHII1 | HKKT-CC | GHKKT-GCC |
| UHII2 | HKLT-CC | GHKLT-GCC |
| UHII3 | HKSL-CC | GHKSL-GCC |
| UHII4 | HKSS-CC | GHKSS-GCC |
| UHII5 | HKWS-CC | GHKWS-GCC |


**Table 8.** RMS of the differences between UHI intensity obtained with meteorological data and GNSS data in
Hong Kong in 2020 (Unit: K).

| Pair of stations | 1-year data | Spring | Summer | Autumn | Winter |
|---|---|---|---|---|---|
| UHII1 | 1.34 | 1.45 | 1.36 | 1.31 | 1.25 |
| UHII2 | 1.03 | 1.12 | 1.03 | 0.96 | 1.03 |
| UHII3 | 1.05 | 1.16 | 1.05 | 0.98 | 0.93 |
| UHII4 | 1.13 | 1.21 | 1.10 | 1.08 | 1.03 |
| UHII5 | 1.23 | 1.32 | 1.24 | 1.14 | 1.25 |


The tables 5 and 6 show the mean UHI intensity of meteorological data and GNSS data in 2020
using 1-year data and data for each season. In all cases, the UHI intensity is the highest during spring,
and the lowest during autumn. The mean UHI intensity in different seasons is less than 0.6 K at the
same station while the mean UHI intensity of one-year data is less than 0.4K. As shown in Figure
8, all RMS of the differences between the UHII obtained with GNSS data and meteorological data
are below 1.5K. In addition, compared with meteorological data, the accuracy of the UHI intensity
is 1.20 K at a 95% confidence level using a full year of GNSS data.
**5. Conclusion**
GNSS radio occultation provides high-precision middle and upper atmospheric parameter profiles
(pressure, water vapor partial pressure and temperature). In this paper, historical radiosonde data
and radio occultation data were used to optimize the ground-based GNSS tomography model to
improve the accuracy of tomography-derived wet refractivity. After obtaining the wet refractivity,



the ERA5 product was used as the initial value, and the search method was used to obtain the best temperature for the wet refractivity. The developed algorithm demonstrated the possibility of using GNSS data to monitor the UHI intensity. Ground-based GNSS data can be used for micro and meso-scale urban climate studies and has the following advantages: 1). the ground-based GNSS tomography technique works in all weather conditions, and its data are widely available as GNSS constellations are designed to cover the earth at all times; 2). GNSS data has a very high temporal resolution and can be processed in real-time or near-real-time.

This study overcomes two major challenges in the algorithm development. The first challenge is the determination of the GNSS tomographic top grid height. Here, we obtained the SGT and FGT based on the RO data and radiosonde products in Hong Kong, and fitted the wet refractivity between FGT and SGT to a multi-order spherical harmonic function based on historical radiosonde and RO products. The height between the earth's surface and SGT was divided into several voxels, and the wet refractivity at the vertex of the voxels was used as an unknown parameter for GNSS tomography. While several voxels are also divided between SGT and FGT, and the wet refractivity at the vertex of voxels was directly obtained based on Eq. (16). Thus, the height of the grid top is decreased, conversely increasing the effective number of the GNSS satellite rays. Moreover, the number of unknowns in GNSS tomography can be reduced, and the accuracy of the tomography results can be improved.

The second challenge is the estimation of temperature from wet refractivity. Based on the relationship between wet refractivity and temperature and water vapor partial pressure, as well as the linear variation of temperature with elevation and the approximate exponential change of water vapor partial pressure with elevation, the optimal search method was used to obtain water vapor partial pressure and temperature from wet refractivity. After selecting five meteorological observing stations inside the city of Hong Kong as urban stations, and a station on an island in Hong Kong as a rural station, we used Eq. (10) to estimate the UHII as the benchmark value of the UHII obtained from GNSS data.

Using the data of 18 stations in Hong Kong in 2020 for a trial calculation, the following conclusions are obtained:

(1) Compared with the radiosonde data, the test results show that the wet refractivity quality obtained by the optimized technique is 16.8% better than that obtained by the traditional technique.

(2) Using the radiosonde product as the benchmark value, the accuracy of the temperature obtained by GNSS data below 600 meters is better than 1.35K.

(3) By solving the RMS of the differences between UHII obtained from GNSS data and meteorological data on the 5 selected locations, it has been shown that, the difference of the UHII obtained from GNSS data and the measured UHII using temperature data in spring and summer is higher than other seasons, because the water vapor content is more abundant in these two seasons. Therefore, the water vapor partial pressure is not accurately calculated in spring and summer. The discrepancies between the HUI estimated by the algorithm and the UHII obtained from meteorological stations can be attributed to the lack of water vapor partial pressure data and GNSS processing. The new algorithm can be used to monitor the diurnal cycle of the UHI.

**Data Availability**



The radiosonde observations (IGRA2, 2022) can be downloaded from the following website:
https://www1.ncdc.noaa.gov/pub/data/igra.html. In addition, the fifth generation of the European
Centre for Medium-Range Weather Forecasts (ECMWF) reanalysis (ERA5,2022) can also be
collected free of charge from https://apps.ecmwf.int/datasets/data/interim-full-daily. WEGC GNSS
RO OPSv5.6 data are supported by the WEGC EOPAC team and are available online (see
https://doi.org/10.25364/WEGC/OPS5.6:2021.1)

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

**Author contributions.**

P. Xia, P. Wei, and S. Ye contributed to the conception of the study;
P. Xia, P. Wei, S. Ye, and M. Guo performed the experiment;
P. Xia, P. Wei, S. Ye, and F. Hu contributed significantly to analysis and manuscript preparation;
P. Xia, P. Wei, and F. Hu performed the data analyses and wrote the manuscript;
P. Xia, S. Ye, and M. Guo helped perform the analysis with constructive discussions.

**Competing interests.**

The authors declared that they have no conflicts of interest to this work.
We declare that we do not have any commercial or associative interest that represents a conflict
of interest in connection with the work submitted.
