# Peer review of "Monitoring Urban Heat Island Intensity with Ground-based GNSS Observations and 2 Space-based Radio Occultation and Radiosonde Historical Data 3 Pengfei Xia1, Wei Peng2, Shirong Ye1, Min Guo3, Fangxin Hu1 4 1GNSS Research Center, Wuh"

_EGUsphere, 2022_

## Author Comment (AC3)

The paper by Pengfei Xia et al. is very interesting, but has also very serious shortcomings in the first part (page 1-5).

The methodology section MUST be rewritten IN DEPTH. Basically:

**Response:** Thank you very much for your invaluable time and great efforts toward our manuscript. We are very appreciative of that all the comments have helped us a lot to improve the manuscript. We have carefully examined each comment and we have tried best to revise and restructure the manuscript based on the valuable comments and suggestions.

- The correct mathematical setting of eq.1 is that of the so-called Radon transform. Please have a look at the literature on this tranform (for example X-ray imaging) and mention some theoretical references.

**Response:** Thank you very much for your valuable comments and suggestions.

Eq.1 has been applied to GNSS meteorology for more than two decades, and we have referred to these literatures (Flores et al., 2000; Troller et al., 2002; Emardson and Webb, 2002; Champollion et al., 2005; Chen and Liu et al., 2014;Dong et al., 2018; Zhang et al., 2020, 2021). Thank you very much for your suggestion, We will do in-depth research on this equation in the future.

Reference

Champollion C, Masson F, Bouin MN, Walpersdorf A, Doerflinger E, Bock O, Van Baelen J. 2005. GPS water vapour tomography: Preliminary results from the ESCOMPTE field experiment. Atmos Res. 74(1-4):253-274.

Chen B, Liu Z. 2014. Voxel-optimized regional water vapor tomography and comparison with radiosonde and numerical weather model. J Geodesy. 88(7):691-703.

Emardson T, Webb FH. 2002. Estimating the motion of atmospheric water vapor using the Global Positioning System. Gps Solut. 6(1-2):58-64.

Dong Z, Jin S. 2018. 3-D water vapor tomography in Wuhan from GPS, BDS and GLONASS observations. Remote Sens-Basel. 10(1).

Flores A, Ruffini G, Rius A. 2000. 4D tropospheric tomography using GPS slant wet

delays. Ann Geophys-Germany. 18(2):223-234.

Troller M, Bürki B, Cocard M, Geiger A, Kahle HG. 2002. 3-D refractivity field from GPS double difference tomography. Geophys Res Lett. 29(24).

Zhang W, Zhang S, Ding N, Zhao Q. 2020. A tropospheric tomography method with a novel height factor model including two parts: Isotropic and anisotropic height factors. Remote Sens-Basel. 12(11).

Zhang W, Zhang S, Chang G, Ding N, Wang X. 2021. A new hybrid observation GNSS tomography method combining the real and virtual inverted signals. J Geodesy. 95(12).

- the decay with respect to altitude of temperature (linear) and water vapor contents (exponential) is not always true.

**Response:** Thank you very much for your commons.

It is actually that the decay with respect to altitude of temperature (linear) and water vapor contents (exponential) is not always true.

According to the statistical value of radiosonde from 2015 to 2020, the proportions of the decay with respect to altitude of temperature (linear) and water vapor pressure (exponential) from surface to 600 m are 57.3% and 66.2%, respectively.

inversion layers are common,as noted by the authirs themselves...but later in the paper.

**Response:** Thank you very much for your commons.

Our main purpose is to obtain the wet refractivity from the surface to 600m elevation based on GNSS tomography method. Although GNSS tomography fails to accurately represent the "inversion layer", the wet refractivity accuracy obtained by the optimized tomography technique is higher than that of the conventional tomography technique.

- even more important, and I would say a major flaw in the method is that its robustness with respect to small variations

**Response:** Thank you very much for your commons.

in the beta coefficient (eq. 7) and *a* and *b* (eq. 8) is not addressed. This MUST be

discussed and robustness established.

**Response:** Thank you very much for your commons.

*β*, *a* and *b* are the experience value that can be determined using radiosonde products. The main purpose of this manuscript is to demonstrate whether the urban heat island intensity can be monitored based on GNSS tomography method. In order to strictly controlled the entire algorithmic flow, then we will try to separate the atmospheric temperature from the wet refractivity using the variational analysis method in the future.

- The use of a "Kalman filtering" as a magic word to invert SWD values as Nw values. Kalman filtering is just another word for a least-squares process. In this particular case, the authors are technically doing a least-squares linear inverse problem. I urge them to have a look at the fundamental papers by Tarantola and Valette around 1980, that can be easily found, and especially the paper "inverse problems = Quest for information".

**Response:** Thank you very much for your commons.

In the tomographic approach, the observation equation is ill-conditioned as satellite signals do not pass through all voxels, causing the non-uniqueness of the tomography solutions. In order to solve this issue, a variety of reconstruction algorithms have been developed. They may be generally grouped into two categories. One is the iterative reconstruction technique (IRT) such as the algebraic reconstruction techniques (ART) (Wen et al., 2010; Bender et al., 2011), the multiplicative algebraic reconstruction techniques (MART) (Stolle et al., 2006; Jin et al., 2008) and the simultaneous iterative reconstruction techniques (SIRT) (Liu et al., 2010). Another is the non-iterative reconstruction technique (NIRT) such as the singular value decomposition technique (SVD) (Flores et al., 2000; Champollion et al., 2005; Notarpietro et al., 2011). In addition, the Kalman filtering (Nilsson andGradinarsky, 2006). So far, it is a very common method to solve GNSS tomography using 'Kalman filtering'(Dong et al., 2018; Ding et al., 2018; Zhao et al., 2020), so we did not describe in detail how to solve the tomographic equation using 'Kalman filtering' in the manuscript.

Reference.

Bender, M., Dick, G., Ge, M., Deng, Z., Wickert, J., Kahle, H. G., Raabe, A. and Tetzlaff, G. 2011.

Development of a GNSS water vapour tomography system using algebraic reconstruction techniques, Adv. Space Res., 47, 1704-1720, doi:10.1016/j.asr.2010.05.034, 2011.

Champollion, C., Masson, F., and Bouinm, N. 2005. GPS water vapour tomography: preliminary results from the ESCOMPTE field experiment, Atmos. Res., 74, 253-274.

Ding N, Zhang SB, Wu SQ, Wang XM, Zhang KF. 2018. Adaptive Node Parameterization for Dynamic Determination of Boundaries and Nodes of GNSS Tomographic Models. Journal of Geophysical Research: Atmospheres. 123(4):1990-2003.

Dong Z, Jin S. 2018. 3-D water vapor tomography in Wuhan from GPS, BDS and GLONASS observations. Remote Sens-Basel. 10(1).

Flores, A., Ruffini, G., and Rius, A. 2000. 4D tropospheric tomography using GPS slant wet delays, Ann. Geophys., 18, 223-234.

Jin, S. G., Luo, O. F., and Park, P. 2008. GPS observations of the ionospheric F2-layer behavior during the 20th November 2003 geomagnetic storm over South Korea, J. Geophys. Res., 82, 883-892.

Liu, S. Z.,Wang, J. X., and Gao, J. Q. 2010. Inversion of ionosphere electron density based on a constrained simultaneous iteration reconstruction technique, IEEE T. Geosci. Remote, 48, 2455-2459.

Nilsson, T. and Gradinarsky, L. 2006. Water vapour tomography using GPS phase observation: simulation results, IEEE Trans. Geosci.Remote Sens., 44, 2927-2941.

Wen, D. B., Liu, S. Z., and Tang, P. Y. 2010. Tomographic reconstruction of ionospheric electron density based on constrained algebraic reconstruction technique, GPS Solut., 14, 251-258.

Zhao Q, Yao W, Yao Y, Li X. 2020. An improved GNSS tropospheric tomography method with the GPT2w model. Gps Solut. 24(2).

By the way (for the authors), are you doing a linearization of the inversion problem around eqs. 5 and 7? How do you weight a priori information, if any? Please add the relevant equations and do not stay in the vague of "Kalman filtering".

**Response:** Thank you very much for your commons.

After obtaining the wet refractivity profile based on the GNSS tomography method, the temperature was estimated by the optimal search method using eqations (6), (7) and (8).

The fifth-generation reanalysis model (ERA5) could provide temperature and water vapor partial pressure, which were selected as the initial values in this study. The flow chart of data processing is as follows:

[Figure]

**Minor points:**

- please add contouring of the topography in Figure 1.

**Response:** Thank you very much for your commons. Figure 1 has added the contouring of the topography.

[Figure]

- please describe in a few sentences what is GNSS RO. Are you using COSMIC-2 data?

**Response:** Thank you very much for your commons. We added the detail of RO events happened in Hong Kong.

| | 2010 |  |  |  | 2011 |  |  |  | 2012 |  |  |  | 2013 |  |  |  | 2014 |  |  |  | 2015 |  |  |  | 2016 |  |  |  | 2017 |  |  |  | 2018 |  |  |  | 2019 |  |  |  |
|---|--|--|--|--|--|--|--|--|--|--|--|--|--|--|--|--|--|--|--|--|--|--|--|--|--|--|--|--|--|--|--|--|--|--|--|--|--|--|--|--|
| | Q1|Q2|Q3|Q4 | Q1|Q2|Q3|Q4 | Q1|Q2|Q3|Q4 | Q1|Q2|Q3|Q4 | Q1|Q2|Q3|Q4 | Q1|Q2|Q3|Q4 | Q1|Q2|Q3|Q4 | Q1|Q2|Q3|Q4 | Q1|Q2|Q3|Q4 | Q1|Q2|Q3|Q4 |
| GRACE | | | | |
| MetOp-A | | | | |
| MetOp-B | | | | |
| MetOp-C | | | | |
| TDX | | | | |
| TSX | | | | |
| COSMIC-1 | | | | |
| COSMIC-2 | | | | |
| SACC | | | | |
| PAZ | | | | |
| Kompasat5 | | | | |

Fig.1. Selected radio occultation products and the corresponding time span. 'Q' means quarterly.

Table 1. Detail of RO events happened in Hong Kong

| | |
|---|---|
| The range of selected RO events | 21.2 °N-23.6 °N; 112.85 °E-115.15 °E |
| Mean mumber of RO events | 1.3/day |
| The type of RO events | post-processed data products |
| The level of RO events | Level2 |

At this point, my recommendation is to reject and resubmit once these major issues have been fixed, or at minimum major revision, to be sure that the second part of the paper is reliable.

**Response:** Thank you very much for your commons.

If there is an inappropriate answer, please put it up again, I am very happy to answer and revise the manuscript again. Thank you very much.

---

## Author Comment (AC4)

After the careful review, my conclusion is the study is not accepted. The further comments are as follows for the considerations.

**Response:** Thank you very much for your invaluable time and great efforts toward our manuscript. We are very appreciative of that all the comments have helped us a lot to improve the manuscript. We have carefully examined each comment and we have tried best to revise and restructure the manuscript based on the valuable comments and suggestions.

In addition, if there is an inappropriate answer, please put it up again, I am very happy to answer and revise the manuscript again. Thank you very much.

1. About tomography, the definition of voxels (volumetric pixels) should be the first important thing. I cannot see any clear definition for it. I mean what the size of voxels used in this study. I suggest that the authors put it at the beginning of tomography section.

   **Response:** Thank you very much for your suggestions. We added the the size of voxels in the vertical direction at the beginning of tomography section.

   "The first stage is from the ground to 1km, which be further divided into 3 layers: the heights of the first two layers are 300m, and the height of the third layer is 400m. The second stage is from 1km to SGT. The grid in this stage is divided into an even vertical height which requires the height of a grid is no less than 400m and not more than 600m. We divide it into (11±1) layers and each layer height is (SGT-1000) / (11±1) m. The last stage is from the SGT to the FGT. The grid in this stage is divided into an even vertical height which requires the height of the grid is no less than 600m, and not greater than 1000m. We divide it into (6±1) layers and each layer height is (FGT-SGT) / (6±1) m."

2. Slant wet delay (SWD) is the beginning of everything about tomography in this manuscript. The authors just simply only applied Niell Mapping Function and without any parameterised strategy. It is highly risky, especially for water vapour (wet delay). The residual item should be in the equation (13) and the residual

should be estimated. Actually, there are at least 3 different kinds of tomography models for 3-D (BIRA, TUW…)and one for 4-D can be used. Applying mapping function only in this manuscript is not enough.

**Response:** Thank you very much for your commons.

Actually, we take into account the influence of the horizontal gradient, equation (13) can be revied as

$$\text{SWD}_h = \text{ZWD}_h \cdot M_w^{\text{Niell}}(e_{min}) + \frac{1}{\sin(\alpha) \cdot \tan(\alpha) + 0.0007}[G_N\cos(e) + G_E\cos(e)]$$

where, $G_N$ is the north-south wet gradient component, and $G_E$ the wet gradient component relative to the east-west direction, $e$ is the satellite azimuth; and α is the satellite cut-off altitude.

3. I suggest that more literature review about tomography is necessary. Atmospheric Measurement Techniques (EGU) and ETH Zurich, Switzerland, JoG are good resources.

**Response:** Thank you very much for your commons. We added the latest literature on improving GNSS tomography in the introduction.

"At present, scholars at home and abroad focus on how to improve GNSS water vapor tomography technology and further improve the accuracy of GNSS tomography. Such as, combining other observation data or reducing the error in signal propagation can also further improve the accuracy of wet delay information (Möller and Landskron, 2019; Heublein et al., 2019). In terms of solving tomographic equations, scholars have analyzed and improved the algebraic reconstruction method to improve the speed and accuracy of solution (Xia et al., 2013; He et al., 2015), and some scholars have also used the compressed sensing method to solve the formula (Heublein et al., 2019). A large number of experiments have proved as well that high-precision prior information can help improve the vertical and horizontal constraints and improve the accuracy of tomography (Chen and Liu et al., 2016; Benevides et al., 2018; Xia et al., 2018)."

4. The authors used wet refractivity from radiosonde and GPS RO to retrieve temperatures as the constrain in the study. GPS RO also provide temperature profiles. Is any reason the authors did not use the GPS RO temperature profiles directly.

   **Response:** Thank you very much for your commons.

   The satellite-based GPS RO data have provided continuous temperature and pressure measurements as a function of height in the upper troposphere with high accuracy, high vertical resolution. However, the temperature profiles provided by the GPS RO has poor accuracy in the lower troposphere and cannot be used as a benchmark vaule.

5. Line 328 and 329, "….ERA5 could provide….., which were selected as the initial values in this study…..", GPS RO and radiosondes have already been assimilated into ERA5. However, the authors used ERA5 as the first guest for the wet profiles constrained by GPS RO. Why don't the authors use ERA5 as the constrain directly for first and second grid tops?

   **Response:** Thank you very much for your commons.

   Compared to ERA5, both GPS RO data and radiosondes have provided continuous temperature and pressure measurements as a function of height in the upper troposphere with higher accuracy, higher vertical resolution. We think the wet refractivity between the first grid top and the second grid top can be betther fited by selecting the RO and radiosonde products than ERA5. In addition, we will refer to your suggestions, and we will use ERA5 as the constrain directly for first and second grid tops in the future and compared it to our results. Thank you very much.

6. Line 108 and 109, "Though the Gauss weighted method (Song, 2004) can be used for the horizontal direction,…………". This strategy is only applied when the horizontal distribution of water vapour is stable. However, the weather (or climate) is the non-linear and complex system. I think the method is only suited for the simulation.

**Response:** Thank you very much for your commons.

Gaussian distance weighting method is the horizontal constraint method chosen by many scholars who do GNSS tomography research (Ye et al., 2016; Zhao et al., 2019; Zhang et al.,2020). Indeed, this method has some shortcomings, such as it only applied when the horizontal distribution of water vapour is stable. Our next step will focus on finding a more efficient horizontal constraint method to improve the results of GNSS tomography, thank you very much for your suggestion.

Reference.

Ye S, Xia P, Cai C. 2016. Optimization of GPS water vapor tomography technique with radiosonde and COSMIC historical data. Ann Geophys-Germany. 34(9):789-799.

Zhang W, Zhang S, Ding N, Zhao Q. 2020. A tropospheric tomography method with a novel height factor model including two parts: Isotropic and anisotropic height factors. Remote Sens-Basel. 12(11).

Zhao Q, Yao Y, Yao W, Zhang S. 2019. GNSS-derived PWV and comparison with radiosonde and ECMWF ERA-Interim data over mainland China. J Atmos Sol-Terr Phy. 182:85-92.

7. If the GPS RO is applied to this study, the number and the details of RO events happened in Hong Kong should be presented as the table in the manuscript and the types of RO data (used Level) as well.

   **Response:** Thank you very much for your commons. We added the detail of RO events happened in Hong Kong.

| | 2010 | | | | 2011 | | | | 2012 | | | | 2013 | | | | 2014 | | | | 2015 | | | | 2016 | | | | 2017 | | | | 2018 | | | | 2019 | | | |
|---|---|---|---|---|---|---|---|---|---|---|---|---|---|---|---|---|---|---|---|---|---|---|---|---|---|---|---|---|---|---|---|---|---|---|---|---|---|---|---|---|
| | Q1 | Q2 | Q3 | Q4 | Q1 | Q2 | Q3 | Q4 | Q1 | Q2 | Q3 | Q4 | Q1 | Q2 | Q3 | Q4 | Q1 | Q2 | Q3 | Q4 | Q1 | Q2 | Q3 | Q4 | Q1 | Q2 | Q3 | Q4 | Q1 | Q2 | Q3 | Q4 | Q1 | Q2 | Q3 | Q4 | Q1 | Q2 | Q3 | Q4 |
| GRACE | | | | | | | | | | | | | | | | | | | | | | | | | | | | | | | | | | | | | | | | |
| MetOp-A | | | | | | | | | | | | | | | | | | | | | | | | | | | | | | | | | | | | | | | | |
| MetOp-B | | | | | | | | | | | | | | | | | | | | | | | | | | | | | | | | | | | | | | | | |
| MetOp-C | | | | | | | | | | | | | | | | | | | | | | | | | | | | | | | | | | | | | | | | |
| TDX | | | | | | | | | | | | | | | | | | | | | | | | | | | | | | | | | | | | | | | | |
| TSX | | | | | | | | | | | | | | | | | | | | | | | | | | | | | | | | | | | | | | | | |
| COSMIC-1 | | | | | | | | | | | | | | | | | | | | | | | | | | | | | | | | | | | | | | | | |
| COSMIC-2 | | | | | | | | | | | | | | | | | | | | | | | | | | | | | | | | | | | | | | | | |
| SACC | | | | | | | | | | | | | | | | | | | | | | | | | | | | | | | | | | | | | | | | |
| PAZ | | | | | | | | | | | | | | | | | | | | | | | | | | | | | | | | | | | | | | | | |
| Kompasat5 | | | | | | | | | | | | | | | | | | | | | | | | | | | | | | | | | | | | | | | | |

Fig.1. Selected radio occultation products and the corresponding time span. 'Q' means quarterly.

Table 1. Detail of RO events happened in Hong Kong

| | |
|---|---|
| The range of selected RO events | 21.2 °N-23.6 °N; 112.85 °E-115.15 °E |
| Mean mumber of RO events | 1.3/day |
| The type of RO events | post-processed data products |
| The level of RO events | Level2 |

8. Both of CDAAC 2.0 (UCAR, USA) and ROM SAF (EUMETSAT) also provide GPS RO profiles. Is it the same with WEGC OPSv5.6 ? If not, I suggest that the authors provide the reason for using WEGC OPSv5.6.

**Response:** Thank you very much for your commons.The most serious problem is a lack of quality control for CDAAC 2.0 (UCAR, USA). The CDAAC distributes RO data for all RO missions to date—except for the FenYung satellites, whose data is restricted—but the data hosted at the CDAAC is extremely heterogeneous, having been processed with different versions of the RO retrieval system at UCAR. WEGC OPSv5.6 overcomes this shortcoming (Meng et al. 2021).

Reference.

Meng, L.Y., Liu, J., Tarasick, D.W., Randel, W.J., Steiner, A.K., Wilhelmsen, H., Wang, L. and Haimberger, L. 2021. Continuous rise of the tropopause in the northern hemisphere over 1980-2020. Sci. Adv., 7: 1-9. https://doi/10.1126/sciadv.abi8065

9. The ZTD of ground based GPS stations are retrieved from GAMIT software package. I suggest the authors provide the used strategy in the manuscript for getting ZWD from GAMIT's ZTD. Theoretically, the in situ weather stations are necessary then you just can get precise ZWD from ZTD.

   **Response:** Thank you very much for your commons.

   We used 'GPT3+ Saastamoinen' model to correct the Zenith Hydrostatic Delay (ZHD), ZWD can then be obtained by removing ZHD from ZTD.

10. I suggest the authors do the comparison with MODIS data for heat island topic.

    **Response:** Thank you very much for your suggestions.

    It is a valuable suggestion. The temperature obtained from MODIS data is usually the temperature at the earth's surface or the top floor of a building, while the temperature obtained from the GNSS tomography is the temperature at the elevation of the GNSS station. The spatial and temporal resolutions of the two products are not the same. Our next step is to unify the spatio-temporal resolution of the two products and compare them.

11. I suggest that the authors put "tomography" in the title of manuscript

    **Response:** Thank you very much for your suggestions. The title of manuscript will be revised as "Monitoring Urban Heat Island Intensity based on GNSS tomography technique".